# Indigenous Community-Led Programs to Address Food and Water Security: Protocol for a Systematic Review

**DOI:** 10.3390/ijerph18126366

**Published:** 2021-06-11

**Authors:** Ashleigh Chanel Hart, Emalie Rosewarne, Wendy Spencer, Ruth McCausland, Greg Leslie, Janani Shanthosh, Christine Corby, Keziah Bennett-Brook, Jacqui Webster

**Affiliations:** 1The George Institute for Global Health, Sydney, NSW 2042, Australia; erosewarne@georgeinstitute.org.au (E.R.); jshanthosh@georgeinstitute.org.au (J.S.); kbennett-brook@georgeinstitute.org.au (K.B.-B.); jwebster@georgeinstitute.org.au (J.W.); 2Dharriwaa Elders Group, Walgett, NSW 2832, Australia; wendy@yuwayangarrali.org.au; 3Institute for Global Development, The University of New South Wales, Sydney, NSW 2052, Australia; ruth.mccausland@unsw.edu.au; 4UNSW Global Water Institute, The University of New South Wales, Sydney, NSW 2207, Australia; g.leslie@unsw.edu.au; 5Australian Human Rights Institute, Sydney, NSW 2052, Australia; 6Walgett Aboriginal Medical Service, Walgett, NSW 2832, Australia; ChristineC@walgettams.com.au

**Keywords:** indigenous, community-led, food and water security, food insecurity, water insecurity, nutrition, community food and water, systematic review, protocol

## Abstract

The connection between indigenous peoples and Country (a multidimensional concept including land and water) enabled communities to thrive and survive over millennia. This has been eroded by colonisation, dispossession and increasing food and water insecurity due to climate change and supply constraints. Globally, indigenous peoples experience a disproportionate burden of chronic disease and poor nutrition is a major risk factor. Indigenous leaders have been advocating for community-led solutions. The primary aim of this systematic review is to determine what community-led programs have been undertaken to address food and/or water security globally. A comprehensive search of peer-reviewed literature will be performed in EMBASE, CINAHL, PsycINFO, PubMed, Scopus, LILACs, Informit and Business Source Premier. The grey literature search will include grey literature databases, customised Google search engines, targeted websites, and consultation with experts. The search strategy will consist of four concepts, combined as follows: (1) indigenous peoples AND (2) community program AND (3) food security OR (4) water security. Covidence will be used for study screening and data extraction by two authors. A deductive thematic analysis using indigenous-informed methodologies will be used to synthesise data. This review seeks to provide insight on models and mechanisms to encourage action and metrics for quantifying success of indigenous community-led programs to improve food and water security.

## 1. Introduction

The connection between indigenous peoples and country (a multidimensional concept including land and water) has enabled communities to thrive and survive over millennia [1]. This has been eroded by colonisation, dispossession and increasing food and water insecurity due to climate change and supply constraints [2,3,4]. Globally, indigenous peoples experience a disproportionately higher burden of death and disease than nonindigenous peoples [5,6,7]. Poor nutrition is a major risk factor for diet-related noncommunicable diseases (NCDs) [8,9]. Food insecurity is associated with worsening nutrition and increased NCDs [10]. Increased frequency of extreme weather events as a result of climate change, and more recent crisis events, including the current COVID-19 pandemic, have affected food and water security [2,11]. Local food and water systems are influenced by various factors, including infrastructural developments and disruptions, which can enable or hinder community food security and nutrition experienced [12]. In Australia, previous government-led attempts to improve food security and/or nutrition have had limited impact on indigenous communities, including the ‘Close the Gap’ campaign, the ‘National Aboriginal and Torres Strait Islander Peoples in Australia Nutrition Strategy and Action Plan’, the ‘Measure Up’ healthy-eating social marketing campaign, and the ‘Outback Stores’ affordable food program [13]. Despite these programs acknowledging that poor nutrition underlies the disproportionate burden of chronic disease that Aboriginal and Torres Strait Islander peoples experience, there has been limited government commitment to policy and program action, inadequate funding, and lack of consultation with, and involvement of, Aboriginal and Torres Strait Islander peoples, which ultimately undermined policy/program effectiveness and sustainability. This highlights the need for alternative indigenous-led and informed approaches and methods to reframe the narrative, create effective programs and achieve sustainable solutions in communities [13]. We respectfully acknowledge the diversity of indigenous peoples worldwide and we use the term ‘indigenous’ collectively and inclusively in this paper while acknowledging that it is viewed by some people as a problematic term.

Indigenous leaders and community-controlled organisations (CCOs) globally have advocated for community-led solutions, manifesting a more pronounced leadership role recently to raise awareness of the asymmetry of food and water security between indigenous and nonindigenous populations amidst the COVID-19 pandemic [14,15]. This advocacy is grounded in indigenous knowledge systems and draws on human rights frameworks and evidence of improved environmental and health outcomes. While dominant approaches to food security focus on securing physical and economic access to sufficient, safe and nutritious food, the Food Sovereignty movement challenges this paradigm and seeks to build alternative food regimes grounded in an understanding of the need to challenge colonial thinking and structures, and decolonise research approaches [16,17]. Enshrined in Article 25 of The Declaration on the Rights of indigenous Peoples [18], Food Sovereignty is a rights-based approach, whereby all people have the right to healthy and culturally appropriate food produced through ecologically sound and sustainable methods [19]. Likewise, a Water Sovereignty approach is based on underlying values whereby land, water, humans, and nature are integrated, and indigenous knowledge systems are prioritised [20]. In cases where this approach has been adopted, and indigenous communities are directly engaged in management of rivers and catchments, better environmental and health outcomes are consistently experienced [21]. Thus, community-led programs are crucial for sustainable solutions for indigenous peoples because they use a rights-based and decolonising approach [22], and may also provide an effective model for food and water security efforts more generally.

A stark example of the deleterious effect of food and water insecurity on the health of indigenous peoples is in Walgett, a remote town in New South Wales, Australia, where colonisation, institutional racism, drought, mismanagement of rivers and systemic poverty have had a major negative impact. The Yuwaya Ngarra-li (YN) (‘Vision’) partnership between the Dharriwaa Elders Group (DEG, an Association of Aboriginal Elders in Walgett, Australia) and the University of New South Wales (UNSW) [23,24,25] are coordinating a community-led program, the ‘Walgett Food and Water For Life Program’ (WFW4L) [26]. The aim of WFW4L is to improve nutrition and wellbeing outcomes of the local Aboriginal community through sustainable solutions to food and water insecurity, and developing and evaluating a model for improving outcomes in Aboriginal and Torres Strait Islander communities across Australia [26]. As collaborators on this program, in order to support WFW4L to achieve long-lasting success, the authors seek to understand what community-led programs have been developed and effectively implemented elsewhere. The aim of the planned systematic review is to identify and describe indigenous community-led programs globally that can lead to better nutrition, health and wellbeing outcomes. It is hoped the review findings can make a valuable contribution to the evidence base and inform progressive policymaking to further enable indigenous community-led programs that improve the health and wellbeing of indigenous peoples globally.

### 1.1. Defining Community-Led Programs

Community-led programs are coordinated by leaders, members, or CCOs from within indigenous communities. For the purposes of this review, we extend this definition to include participatory research, cocreation, and codesign research methods where indigenous peoples from within communities have been involved in the development and the implementation of the program being studied [22].

### 1.2. Objectives of the Review

The objective is to conduct a systematic review to address the following questions:what community-led programs have been or are being conducted in indigenous communities globally to improve food and/or water security?what mechanisms and processes were used to facilitate action at community-level?in addition to food and/or water security, what other outcomes were measured (e.g., nutrition, health and wellbeing)?how have previous indigenous community-led programs defined impact and success?what methods are used to monitor and evaluate programs?

## 2. Methods

This systematic review has been registered with the International Prospective Register of Systematic Reviews (PROSPERO) (CRD 42021227614) [27]. We used the Preferred Reporting Items for Systematic Reviews and Meta-Analysis Protocols (PRISMA-P) checklist when writing this protocol [28] and we will conduct this systematic review in line with the PRISMA guidelines [29].

### 2.1. Searches

An electronic literature search will be conducted using the following databases: PubMed, Embase, PsycINFO, CINAHL, Scopus, LILACS, Informit, and Business Source Premier. In addition, the New Zealand Journal of Indigenous Scholarship will be manually searched for nonindexed articles. All peer-reviewed original research articles published before April 2021 will be included. Reference lists of included articles will be scanned for additional relevant resources.

To minimise risk of omitting relevant sources and publication bias, we will search for grey literature using four searching strategies, which will include: (1) grey literature databases e.g., Indigenous Studies Portal, the Indigenous Knowledge Network for Infant, Child and Family Health, and the Australian indigenous HealthInfoNet; (2) customised Google search engines: Google Scholar (the first 300 hits), and Google Advanced Search; (3) targeted websites e.g., The International Work Group for Indigenous Affairs, the United Nations Indigenous Peoples Assistance Facility (Global), the International Food Policy Research Institute, the Food and Agriculture Organization of the United Nations; and (4) consultation with experts: We will contact experts in the field of indigenous health and nutrition research, indigenous water security research, as well as indigenous community stakeholders for published and unpublished reports. Reference lists of included papers will be scanned for additional resources.

### 2.2. Search Strategy and Terms

The search strategy consists of the combination of four key concepts, as follows: (Indigenous Peoples) AND (Community Program) AND (Food Security) OR (Water Security) Key words for Indigenous Peoples were sourced from the RefWorld Organisation website via a thorough country-by-country review of World Directory of Minorities and Indigenous Peoples data [30] and then cross-checked with available country data listed in the International Work Group for Indigenous Affairs website [31]. This was to ensure the integrity of a broad global indigenous population search. Indigenous leaders were also consulted about keywords. Key words for all other concepts were developed via engagement with experts from within the review team.

### 2.3. Article Screening

Studies considered for inclusion are not limited by study design. Screening will be conducted using Covidence, [32] an online data management software for systematic reviews, and will be conducted according to the following inclusion and exclusion criteria.

#### 2.3.1. Inclusion Criteria

Original research articles published in peer-reviewed journals.Grey literature articles published online.Articles with focus on indigenous community-led programs that address food and/or water security.Full text available in English.

#### 2.3.2. Exclusion Criteria

Articles that do not focus on indigenous populations.Articles that do not focus on community-led programs. Furthermore, articles in which there is no comment on the level of inclusion or involvement of indigenous peoples in the development and implementation of programs will be excluded unless there is enough justifiable evidence throughout the full text of their involvement.Articles that do not focus on food and/or water security.Studies conducted on animals.Articles where the full text is not available.

### 2.4. Article Selection

Study selection and reporting will follow the PRISMA-equity guidelines. A 10% subsample of title and abstracts will be screened by two authors to assess potential eligibility and any discrepancies will be reviewed and resolved. The remaining search results will be screened by one author, who will consult the second author if there is any uncertainty. The full text publications of studies considered potentially eligible will be retrieved and screened by two authors. Disagreements on whether to include a study will be resolved through discussion until consensus is reached, and consultation with the research team and collaborators if necessary. Reference lists of included full-text articles will be screened by one author and follow the same selection procedure. Grey literature will be screened with the same procedure by the respective item’s abstract, executive summary, or table of contents, followed by full text screening.

### 2.5. Data Extraction

A customised data extraction template using Microsoft Excel (see Appendix A) will be used to extract and compare relevant variables including study settings, programs, methodologies, results, publication details, country, population size, regional setting, study design, type of program, stated aims/purpose of program, mechanisms (e.g., infrastructural developments), indigenous representation, data collected, outcomes, success of the program, generalisability and limitations. Data extraction for all included articles will be undertaken independently by at least one reviewer with a second reviewer to check. Disagreements will be resolved through discussion, and consultation with the research team and collaborators if necessary.

### 2.6. Quality Assessment of Included Articles

Quality of the studies will be assessed using critical appraisal tools appropriate to their respective study designs, such as those provided by the Joanna Briggs Institute, by at least one reviewer for all included studies, with a second reviewer to check. Disagreements will be resolved through discussion and/or consultation with the research team.

### 2.7. Data Synthesis

Included articles will be grouped and synthesised upon identifying any similarities of studies or study findings found. It is expected that studies will be heterogeneous; therefore, a narrative approach to data synthesis will likely be undertaken. A deductive thematic analysis drawing on decolonising methodologies will be adopted [33,34,35]. This will take account of the relevant guidelines, principles and frameworks for doing research with Aboriginal and Torres Strait Islander Communities [36,37,38,39]. A rights-based perspective that recognises the agency of indigenous peoples, based on the international human rights principles underpinning the United Nations Declaration on the Rights of Indigenous Peoples, will also be incorporated, [18] with a view to reframing the narrative about food and water security in indigenous communities. Aboriginal and Torres Strait Islander researchers within our research team with expertise in decolonising research methods will be involved in the analysis and synthesis of the data. This is expected to provide key themes by which the data can be synthesised; for instance, framing of decolonisation, similarities in community engagement, locality, program types and outcomes. Potential subgroups analysed may be country or jurisdiction and type of program.

### 2.8. Ethics and Dissemination

Ethical clearance is not needed as we are not collecting primary data and only using published data. Communicating our research results in accordance with the AIATSIS code of ethics [37], we will publish this review in a recognised peer-reviewed journal under open access and also in HealthInfoNet. We will also report our findings as a community report to the Walgett community in New South Wales, Australia.

## 3. Limitations

A key limitation we expect in conducting this review is potential for publication bias due to lack of formal or online documentation of indigenous community-led programs, as well as programs that may not have been successful that were not published. The studies included in this global review will be limited to the English language. As decolonising research methods are an area of research being continually developed, particularly among community-led programs, we expect the literature base to be limited.

## 4. Conclusions

Indigenous leaders have advocated for community-led solutions that address food and water security. These are more likely to lead to improved nutrition, health, and wellbeing outcomes than dominant (government-led) approaches. The proposed systematic review will explore what community-led programs have been implemented globally, what models and mechanisms they used to initiate action, and how impact and success is framed and measured. The findings of the review will provide important insights to support community-led programs to address food and water security in indigenous communities and inform progressive policymaking that enables indigenous communities to lead programs that could lead to more sustainable and effective health outcomes.

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
