# Peer review of "Indigenous Community-Led Programs to Address Food and Water Security: Protocol for a Systematic Review"

_ijerph, 2021, doi:10.3390/ijerph18126366_

Round 1

Reviewer 1 Report

I must admit I was excited to see this work, thinking it would be the results of the systematic review rather than the protocol. However, It's well written and steps are clear and the results of the systematic review will be very valuable so i'm really looking forward to reading it.  A few comments in regards to this paper:

  • given this is around First Peoples community led strategies and you're using the terminology 'Indigenous peoples'  i think it's important to note that you're using this term collectively though you respectfully acknowledge diversity / that there are distinct groups etc
  • there does seem to be some exciting traction in the remote food security space in Australia which is being led by Community controlled orgs so it's great that you'll also be consulting with experts in this field to help uncover it all. 

Reviewer 2 Report

The protocol - "Indigenous Community-Led Programs to Address Food and Water Security: Protocol for a Systematic Review" if implemented will be an important protocol  to ensure equity to food and water security amongst the Indigenous peoples. As a global review, the authors may be biased in their interpretation with emphasis on the findings from Australian National Aboriginal and Torres Strait Islander Peoples. However, the inclusion of Indigenous researchers within the team for their insights is a welcome development. For both Indigenous and non-Indigenous peoples, access to quality food and water within a local food system are also influenced by infrastructural developments, income e.t.c.

Apart from the literature searches, other possibilities to engage key stakeholders within the communities through interview may also be explored. Are there efforts to document knowledge that are associated with valuable indigenous crops - such as knowledge of indigenous fauna and flora, their harvesting, processing, storage etc. that promotes sustainability and health? 
